# EMitool: Explainable Multi-Omics Integration for Disease Subtyping

**DOI:** 10.3390/ijms26094268

**Published:** 2025-04-30

**Authors:** Yong Xu, Jun Wu, Chen Chen, Jian Ouyang, Dawei Li, Tieliu Shi

**Affiliations:** 1Center for Bioinformatics and Computational Biology, The Institute of Biomedical Sciences, School of Life Sciences, East China Normal University, Shanghai 200241, China; xuyong0703@163.com (Y.X.); jwu@bio.ecnu.edu.cn (J.W.); nxsfcc@nxnu.edu.cn (C.C.); ouyangjian12@163.com (J.O.); 2Department of Colorectal Surgery, Fudan University Shanghai Cancer Center, 270 Dong’an Road, Shanghai 200032, China; 3Beijing Advanced Innovation Center for Big Data-Based Precision Medicine, Beihang University & Capital Medical University, Beijing 100083, China; 4Key Laboratory of Advanced Theory and Application in Statistics and Data Science-MOE, School of Statistics, East China Normal University, Shanghai 200062, China

**Keywords:** multi-omics integration, disease subtyping, interpretability, biomarkers discovery, precision oncology

## Abstract

Disease subtyping is essential for personalized medicine, enabling tailored treatment strategies based on disease heterogeneity. Advances in high-throughput technologies have led to the rapid accumulation of multi-omics data, driving the development of integration methods for comprehensive disease subtyping. However, existing approaches often lack explainability and fail to establish clear links between subtypes and clinical outcomes. To address these challenges, we developed EMitool, an explainable multi-omics integration tool that leverages a network-based fusion strategy to achieve biologically and clinically relevant disease subtyping without requiring prior clinical information. Using data from 31 cancer types in The Cancer Genome Atlas (TCGA), EMitool demonstrated superior subtyping accuracy compared to eight state-of-the-art methods. It also provides contribution scores for different omics data types, enhancing interpretability. EMitool-derived subtypes exhibited significant associations with the overall survival, pathological stage, tumor mutational burden, immune microenvironment characteristics, and therapeutic responses. Specifically, in kidney renal clear cell carcinoma (KIRC), EMitool identified three distinct subtypes with varying prognoses, immune cell compositions, and drug sensitivities. These findings highlight its potential for biomarker discovery and precision oncology.

## 1. Introduction

Disease stratification is pivotal in precision medicine, aimed at tailoring medical interventions [1]. Identifying disease subtypes and biomarkers is crucial for enhancing diagnostic, predictive, therapeutic, and prognostic precision. Molecular subtyping studies have notably improved cancer classification, showing better correlations with clinical outcomes than traditional methods. Hence, substantial efforts have been dedicated to developing reliable subtyping methods, facilitating the classification of diseases, such as cancer, Parkinson’s, and Alzheimer’s, into uniform groups with enhanced correlation to clinical outcomes [2,3,4].

The rapid progress of high-throughput sequencing allows for a more thorough disease subtyping approach by integrating genomic, transcriptomic, proteomic, and epigenomic data. Multi-omics integration methods have shown to unveil novel biomarkers undetectable through a single omics analysis [5,6,7]. This integration boosts disease subtyping accuracy, offering a holistic view of diseases and their underlying biological processes at multi-omics levels, thus enhancing understanding and treatment [8,9,10].

Given the importance of integrating multiple omics data in understanding complex diseases, various methodological approaches have emerged.

**Network-based methods**, such as Similarity Network Fusion (SNF) [11], construct a comprehensive patient similarity network by iteratively propagating affinity matrices. While SNF effectively integrates diverse omics data and captures sample relationships, it requires multiple iterations for convergence, leading to high computational complexity in large-scale datasets.

**Clustering ensemble methods**, like PINSPlus [12], leverage perturbation-based clustering techniques to enhance robustness against noise and variability in different omics layers. Although it is computationally efficient, its prognostic performance in disease subtyping remains suboptimal.

**Graph-based and local similarity methods**, including NEMO [13], leverage local neighborhood information to cluster samples without requiring the imputation of missing data. However, NEMO assumes that each sample pair has at least one shared omics measurement, which may limit its applicability in incomplete datasets.

**Bayesian and probabilistic models**, such as iClusterPlus [14], apply statistical inference to jointly model multi-omics data. While iClusterPlus is widely used due to its mature implementation, it suffers from high computational complexity and resource consumption. Additionally, its prognostic performance in disease subtyping is often suboptimal.

**Affinity propagation-based approaches**, such as MANAclust [15], enable the automatic selection of relevant features and the optimal number of clusters. However, the resulting subtypes may lack biological significance due to excessive fragmentation.

**Matrix factorization approaches**, including integrative Non-negative Matrix Factorization (intNMF) [16] and LRAcluster [17], decompose multi-omics data into latent components that capture shared structures across omics layers. While these methods are distribution-agnostic and adaptable to various data types, they often suffer from computational intensity and sensitivity to initialization, which may lead to inconsistent clustering results.

**Consensus-based integration techniques**, including CIMLR [18], achieve efficient large-scale data processing and rapid convergence. Nevertheless, the lack of interpretability in the final clustering results limits its clinical applicability.

Despite the availability of various methodologies for integrating multi-omics data, which have demonstrated promising outcomes in clinical prognosis and treatment, these approaches are constrained by challenges related to the interpretability of the individual contributions of each omics type to the integration process, along with subsequent limitations in discerning subtyping results. In this study, we developed an explainable multi-omics integration method (EMitool https://github.com/megasci-lab/EMitool (accessed on 2 April 2025)) that leverages a weighted nearest neighbor algorithm to integrate multi-omics data in a data-driven and interpretable manner. Unlike existing black-box approaches, EMitool assigns explicit weights to each omics type by evaluating the predictive power of within-omics and cross-omics similarity. This allows for the transparent quantification of the relative contribution of each omics layer in defining patient subtypes. By applying EMitool to 31 cancer types from The Cancer Genome Atlas (TCGA), we demonstrate its superior accuracy and clinical relevance, including biomarker identification, immune microenvironment characterization, and treatment recommendations.

## 2. Results

### 2.1. Overview of EMitool for Disease Subtyping

EMitool effectively integrates multiple data types corresponding to a consistent sample set (e.g., patients) (Figure 1A), employing a robust and explainable iterative network fusion strategy (see Materials and Methods). EMitool initially constructs a k-nearest neighbor (KNN) graph (Figure 1B) and a patient-specific similarity matrix for each type of data simultaneously. The KNN graphs are then employed to facilitate within and cross-type predictions (Figure 1C). Subsequently, the similarity between the actual and predicted values is calculated and transformed into a weight matrix, which serves as an indicator of the explainability of each omics type (Figure 1D). The patient-specific similarity matrices from various data types are then consolidated based on this weight matrix (Figure 1E). Lastly, a consensus clustering method is employed to divide patients into distinct subtypes, using the integrated patient-specific similarity matrix as a basis (Figure 1F). These subtypes were then subjected to downstream analysis, including a prognosis assessment, biomarker identification, and drug response analysis.

### 2.2. Comparative Analysis of EMitool and Eight Advanced Methods in Cancer Subtyping

It has been reported in the literature that the integration of mRNA, methylation, and miRNA omics data can achieve the most optimal results in integrated molecular subtyping [19]. Therefore, we conducted a comparative study of EMitool and eight other advanced methods, which also integrate these three types of omics data (IntNMF, LRAcluster, CIMLR, NEMO, MANAclust, iClusterplus, PINSPlus, and SNF), across 31 distinct cancer types from TCGA. Consistent with a previously established benchmarking approach, we assessed performance based on the significance of the difference in overall survival (OS) times among various subtypes and two cluster validity indices, specifically the Davies–Bouldin Index (DBI) and the Calinski–Harabaz Index (CHI).

In order to ensure a fair comparison, we applied the same preprocessing uniformly across all datasets for the 31 cancer types (see Materials and Methods). The top three tools in overall subtyping performance are EMitool, NEMO, and SNF (Figure 2A; Appendix A). The results indicated that EMitool successfully categorized patients into distinct groups with significantly different overall survival (OS) times (*p*-value < 0.05) in 22 out of the 31 cancer types. Similarly, SNF achieved significant patient groupings in 20 out of the 31 cancer types, while NEMO yielded significant subtyping results in 18 out of the 31 cancer types (Figure 2A; Appendix A). Notably, all methods produced significant subtyping results for the KIRC dataset. Among them, we further compared the subtyping outcomes of the top three tools—EMitool, NEMO, and SNF—for the KIRC dataset.

Furthermore, EMitool outperformed SNF by producing significant subtyping for the KIRC dataset. EMitool classified the KIRC patients into three groups, whereas SNF divided them into four (Figure 2B). Upon further analysis, we found that two subtypes generated by SNF were merged by EMitool, and one dominant subtype corresponded to two different subtypes was identified by EMitool (Figure 2B). Additionally, when comparing the results of EMitool and NEMO, we found that the C3 and C5 subtypes identified by NEMO were merged by EMitool (Figure 2B). Moreover, the C3 subtype identified by SNF had an insufficient sample size of only four samples, and the C3 and C5 subtypes identified by NEMO had similarly low sample sizes, with only three and four samples, respectively (Appendix A). This makes these classifications biologically insignificant, as the limited number of samples is insufficient to ensure the reliability and representativeness of these subtypes (Appendix A). Beyond the clinical relevance of the subtyping, we also assessed the performance of EMitool and various baseline tools using two clustering evaluation metrics, which are DBI and CHI. A smaller DBI indicates superior clustering, as it reflects well-separated clusters, while a larger CHI signifies better clustering due to dense and well-defined clusters. The evaluation results demonstrated that EMitool outperformed NEMO and SNF for most cancer types (Figure 2C,D).

Finally, we showed that integrating multiple types of omics data can produce superior clinical subtyping results compared to using only a single type of omics data (Figure 2E; Appendix A). We observed significant differences in the subtyping results when using different types of omics data, with the integrated result showing higher consistence to the results obtained using mRNA data (Figure 2F).

### 2.3. Clinical Significance of EMitool’s Subtyping Results

To elucidate the clinical implications, we conducted a comparative analysis of the clinical characteristics among the patients classified into different subtypes by EMitool (Figure 3). Using the KIRC dataset, which encompasses 317 patients with paired mRNA, DNA methylation, and miRNA data, as a representative example, EMitool stratified these patients into three distinct subtypes, specifically C1, C2, and C3. These subtypes comprised of 36, 124, and 157 patients, respectively (Figure 3A). The explainability of different omics is illustrated in Figure 3B, suggesting that miRNA expression plays a dominant role in the C1 subtype, while DNA methylation alterations do not appear to play a significant role in any of the subtypes. We next evaluated the enrichment of the pathological stage in each subtype. The results showed that the patients classified as C2 subtype, who have the worst prognosis, were significantly enriched in the stage III and stage IV (Fisher’s test, *p*-value < 0.05, Figure 3C). Specially, C2 patients were enriched in the T3/4, M1, and G4 stage (Fisher’s test, *p*-value < 0.05; Figure 3D–F).

The Tumor Mutational Burden (TMB) is a measure of the number of mutations in a tumor’s genome, which has been validated as a potential biomarker for predicting the response to immunotherapy in many cancers [20,21]. Nonetheless, the prognostic role of the TMB is multifaceted and can exhibit variation contingent upon the specific cancer type. In our study, we conducted an exploration of TMB alterations across diverse subtypes. Our observations revealed that the TMB in the C2 cohort was significantly elevated compared to that in the C1 cohort (Figure 3G). However, no significant differences were identified between the C2 and C3 cohorts, despite the divergence in their prognoses. We further examined the genes exhibiting mutations. In total, 1253, 4053, and 4639 genes were found to be mutated in the C1, C2, and C3 cohorts, respectively. Compared to the 908 mutated genes associated with the KIRC prognosis, as determined based on univariate analysis, we found that the majority of prognostic genes were mutated in the C2 cohort as opposed to the C3 cohort (Figure 3H). This suggested that the TMB, particularly based on prognostic genes, could be a critical factor in determining the KIRC prognosis.

### 2.4. Immune Microenvironment Analysis Across Different Subtypes

Considering that the immune microenvironment is crucial for cancer therapy, we further investigated the immune microenvironment across different subtypes. The proportions of ten distinct types of immune cells were estimated utilizing gene signature scores (Figure 4A), which were computed based on the ranking of the normalized expression of the characteristic signature genes corresponding to each cell type [22]. The Cox multivariate analysis results revealed that the proportion of Stroma, Treg, CD4+, and cDC2 cells were significantly associated with the KIRC prognosis (Figure 4B). Particularly, our results showed that a high proportion of stromal and Treg cells was associated with a poorer prognosis in KIRC. Both types of cells were reported to be linked to worse disease-free survival in several types of cancer, such as colon cancer, colorectal cancer, and endometrial cancer [23,24]. CD4+ cells play a role in regulating tumor growth and metastasis, impacting cancer prognosis and treatment outcomes [25,26]. Prior studies have shown that a high proportion of CD4+ cells is associated with better prognosis, consistent with our observations in KIRC patients [27,28]. cDC2 cells are crucial for the immune response against cancer, as they present antigens and activate T cells. However, their influence on cancer prognosis varies depending on the cancer type and other factors [29]. Our findings indicate that an increased proportion of cDC2 cells may positively influence the prognosis of KIRC patients. The subtype C2, associated with the worst prognosis, exhibits the highest proportion of Treg cells, while the subtype C1, associated with the best prognosis, exhibits the lowest proportion of Treg cells, suggesting that the difference in prognosis in KIRC might be attributed to the differences in Treg cell contents (Figure 3A andFigure 4D). There was a significant correlation between T cells and Treg cells, as expected since Treg cells are a subset of T cells (Figure 4C). Treg cells also exhibit a significant positive correlation with myeloid cells and macrophages, but these two cell types have no direct relationship with the KIRC prognosis (Figure 4B,C). However, there was a notable difference in the myeloid cell proportion among KIRC subtypes (Figure 4E). Literature suggests that myeloid-derived suppressor cells (MDSCs) are linked to poor prognosis in solid tumors, positively correlated with cancer stage, and drive T reg cell expansion [24]. Thus, myeloid and T reg cells may have synergistic effects on KIRC prognosis, implying an interplay between these populations in the tumor immune microenvironment across KIRC patterns.

### 2.5. Subtype-Specific Biomarker Identification

Given the poorer prognosis of patients in the C2 subtype, we focused on identifying subtype-specific biomarkers for this group. We utilized miRNA data to illustrate the biomarker-identification process. Univariate Cox regression analysis revealed that 52 miRNAs were significantly associated with prognosis (adjusted *p*-value < 0.01). A further analysis of these 52 miRNAs across the three subtypes showed significant differential expression of 50 miRNAs (Kruskal–Wallis test, adjusted *p*-value < 0.01; Appendix A). The top 10 miRNAs with the most significant differences were extracted to display the results of the univariate Cox regression analysis (Figure 5A) and expression differences among the subtypes (Figure 5B). The differential gene expression analysis between C2 subtype patients and other subtypes identified 281 genes, including 126 upregulated and 155 downregulated genes (Figure 5C). The top 10 miRNAs with the most significant differences were subjected to target gene prediction using the TargetScan, EVmiRNA, and miRWalk databases. The predicted target genes were intersected with the differentially expressed genes in subtype C2 to identify high-confidence genes targeted by each miRNA (Figure 5D; Appendix A). For example, an analysis revealed a significant negative correlation between IMPA2 and hsa-mir-21 (Appendix A), with IMPA2 significantly underexpressed in subtype-C2 patients (Figure 5E). Previous studies have shown that downregulation of the IMPA2 gene inhibits autophagy initiation in renal cell carcinoma, facilitating the proliferation and migration of renal cell carcinoma cells [30]. Thus, it is inferred that hsa-mir-21 may modulate the reduced expression of the IMPA2 gene in the C2 subtype, promoting cancer cell proliferation and migration in this subtype.

Furthermore, we conducted gene set variation analysis (GSVA) using the 50 cancer hallmark gene sets from the MSigDB database [31] to estimate the enrichment scores of these gene sets in each sample. The Kruskal–Wallis test revealed that 41 of these gene sets showed significant variations in enrichment scores among different subtypes (Figure 5F). The top three gene sets with the most significant differences between subtypes were HALLMARK_IL6_JAK_STAT3_SIGNALING, HALLMARK_ALLOGRAFT_REJECTION, and HALLMARK_INFLAMMATORY_RESPONSE (Figure 5F). The enrichment scores of these three gene sets in the C2 subtype were significantly higher compared to other subtypes (Appendix A). This suggests that the C2 subtype, which has the poorest prognosis, exhibits higher levels of cell proliferation and inflammatory response. Furthermore, the upregulation of PCNA and MKI67 genes, associated with tumor cell proliferation, was observed in the C2 subtype, indicating that patients with short-term survival may exhibit aberrant dysprogression of cell cycles and increased tumor cell proliferation (Appendix A).

### 2.6. Drug Response Analysis for the Subtypes

To further illustrate the clinical significance of the subtyping results, we performed a drug response analysis for each subtype (see Section 4). We first collected all targeted drugs used to treat kidney cancer from the Drugbank database [32], along with their target genes, considering only drugs with mechanisms of action as inhibitors or antagonists. Furthermore, we assessed the dispersion of target gene expression across the C1, C2, and C3 subtypes. We identified 21 target genes that were significantly differentially expressed across these subtypes (Appendix A). These genes include *FLT1*, *KDR*, *FLT4*, *BRAF*, *TOP2A*, *FGFR3*, *DHFR*, *PDCD1*, *RAF1*, *CSF1R*, *RET*, *ITK*, *KIT*, *MET*, *PDGFRA*, *SH2B3*, *FGF1*, *FGFR2*, *FGFR1*, *PDGFRB*, and *FLT3*. Since we only considered drugs with mechanisms of action as inhibitors or antagonists, for each subtype, a drug is recommended if its target genes were upregulated. For instance, the high expression level of *FLT4* in C1 suggests the potential usage of *FLT4* inhibitors, such as Sunitinib, Sorafenib, Axitinib, and Lenvatinib drugs. Similarly, the high expression level of TOP2A in C2 suggests the potential usage of TOP2A inhibitors, such as Doxorubicin drugs. The full recommended drug list for each subtype is listed in Appendix A.

## 3. Discussion 

In this study, we introduced a novel multi-omics integration approach designed to facilitate explainable disease subtyping. We rigorously validated the robustness of our method through a comprehensive comparative analysis against eight state-of-the-art techniques. Furthermore, we demonstrated the clinical relevance of the subtyping results obtained using our approach. Leveraging similarity network fusion, our method exhibits considerable potential for broad application. Notably, it transcends multi-omics data integration, offering a versatile framework for integrating multimodal datasets across diverse domains, including single-cell data, microbiome data, electronic medical records, and medical imaging. This adaptability underscores our method’s significant contribution to multimodal data integration in the current era of big data.

A key component of EMitool is the computation of a comprehensive similarity matrix, which is derived by comparing the observed and predicted values across both within- and cross-omics domains. Although this approach is grounded in well-established mathematical measures (e.g., exponential kernels of Euclidean distance), future studies could further explore the biological basis underlying the computed similarity values, providing deeper insights into the connections between observed and predicted omics-level patient similarities.

Despite the robust performance of our method in multi-omics data integration, certain limitations warrant consideration. Firstly, the number of neighbors (K) in the k-nearest neighbors (KNN) computation, a crucial parameter, was set to a default value of K = 20. While dynamic adjustment (e.g., K = 5, 10, 20, or 30) is feasible, future investigations should explore the impact of varying K values. Secondly, determining the optimal number of disease subtypes remains a significant challenge, particularly in the absence of established gold standards. Further research is needed to refine strategies for identifying clinically relevant subtype numbers. Additionally, while our method provides subtype-level explainability, enhancing the granularity of these results is an area for future development. The pursuit of more nuanced and detailed subtype characterization will further amplify the method’s utility in precision medicine.

Although this study primarily focuses on transcriptomics, DNA methylation, and miRNA data, EMitool’s framework is highly adaptable to other omics layers, including proteomics and metabolomics. Given that these data types often exhibit high sparsity and batch effects, future studies will explore their integration within EMitool to enhance its applicability in broader biomedical contexts.

In future work, we plan to incorporate more advanced deep learning techniques—such as graph representation learning—to further enhance the predictive power of our multi-omics integration model. By leveraging graph convolutional networks and other graph-based learning architectures, we aim to capture the complex relationships and interactions among biological entities more accurately, ultimately improving both the accuracy and interpretability of disease subtyping results [33,34].

## 4. Materials and Methods

### 4.1. Data Processing and Normalization

For the integrative analysis of mRNA, methylation, and miRNA, we downloaded level-three data pertaining to 31 distinct cancers from The Cancer Genome Atlas (TCGA). Corresponding clinical data were also sourced from the same platform. Methylation data for various diseases were selected from either 27k or 450k datasets, contingent on the availability of a substantial number of samples. Furthermore, the methylation features were specifically selected from promoter regions, with feature selection based on the 27,578 CpG sites detected using the Illumina HumanMethylation27 BeadChip. For the 450K dataset, we included only these 27,578 CpG sites to ensure data consistency and comparability. In this study, we addressed the issue of differing sample sizes across omics types by employing a strategy of intersecting samples, ensuring that only samples present across all omics datasets were included in the analysis. Detailed information about each data type is presented in Appendix A.

To mitigate the issue of varying patient numbers across omics datasets, we applied an intersection strategy, retaining only patients present across all three data types. Two preprocessing steps were applied, which are outlier removal and normalization. A patient was excluded from consideration if more than 20% of its data were missing for a particular data type. Similarly, any biological feature (such as mRNA expression) that had more than 20% missing values across patients was filtered out. After that, the expression profiles were normalized as follows:$$\tilde{f}=\frac{f-E\left(f\right)}{\sqrt{Var\left(f\right)}}$$
where f is the expression of a biological feature, $\tilde{f}$ is the normalized value, and $E\left(f\right)$ and $Var\left(f\right)$ represent the empirical mean and variance of f, respectively.

However, due to the requirement that input data for IntNMF must be non-negative, we applied the default preprocessing strategy of IntNMF to ensure compatibility with this method. Specifically, the values of each dataset were shifted to the positive direction by adding the absolute value of the smallest negative number. Furthermore, to make the magnitudes comparable across different datasets, each dataset was rescaled by dividing by its maximum value, ensuring that all values were within the range of 0 to 1. This preprocessing approach was adopted to maintain the integrity of the IntNMF algorithm while ensuring a fair comparison with other methods.

### 4.2. Explainable Multi-Omics Data Integration

The integration procedure implemented in EMitool consolidates various types of patient-specific multimodal data into a unified representation, providing a comprehensive view of patient information. Since no single omics dataset can entirely capture patient similarity at the clinical level, cross-omics predictions were performed to incorporate complementary biological insights, thereby enhancing the generated similarity matrix.

We first calculated the affinity matrix for each omics type using the *affnityMatrix* function from the “SNFtool” R package (version 2.3.1) [11] with default parameters. For example, the example matrix for mRNA data was denoted as $A_{r}=\left[a_{r,\;i,j}\right]$ for mRNA, and for DNA methylation data, it was denoted as $A_{m}=\left[a_{m,\;i,j}\right]$. Here, $a_{r,\;i,j}$ and $a_{m,\;i,j}$ represent the similarity of patient i and j based on their mRNA and DNA methylation profiles, respectively.

Subsequently, a patient-wise k-nearest neighbor (KNN) graph was constructed for each omics dataset by calculating Euclidean distances between normalized patient feature profiles. After that, we applied a prediction strategy, analogous to that described in Hao et al.’s study [35], to perform both within omics and cross-omics predictions.

For the within-omics prediction, the feature profile of a target patient was predicted using the average profile of its k-nearest neighbors within the same omics type. For instance, taking the mRNA profile as an example, the within-omics prediction for patient i, denoted as ${\hat{r}}_{i}$, can be estimated as follows:$${\hat{r}}_{r,i}=\sum_{j\epsilon K_{r,i}}\frac{r_{j}}{\left|K_{r,\;i}\right|},$$
where $K_{r,i}$ is the set of k nearest neighbors of patient i obtained from the KNN graphs generated using the mRNA profile. Similarly, the within-omics prediction for the DNA methylation data can be calculated as follows:$${\hat{m}}_{m,i}=\sum_{j\epsilon K_{m,i}}\frac{m_{j}}{\left|K_{m,\;i}\right|},$$
where $K_{m,i}$ is the set of k nearest neighbors of patient i obtained from the KNN graphs generated using the DNA methylation profile.

For the cross-omics prediction, the feature profile of a target patient was predicted using the average profile of its nearest neighbors from a different omics type. For example, the mRNA-based gene expression of a patient can be predicted from the average gene expression of its DNA-methylation-based nearest neighbors, or vice versa. Hence, the prediction of the mRNA profile using the DNA methylation profile can be presented as ${\hat{r}}_{m,\;i}=\sum_{j\epsilon K_{m,i}}r_{j}/\left|K_{m,\;i}\right|$, and the prediction of the DNA methylation profile using the mRNA profile can be presented as ${\hat{m}}_{r,\;i}=\sum_{j\epsilon K_{r,i}}m_{j}/\left|K_{r,\;i}\right|$.

Similarity between the observed and predicted values (within- or cross-omics) was quantified using an exponential kernel of the Euclidean distance. For instance, the similarity between the mRNA-based prediction and the observed mRNA profile for patient i, denoted as $\theta_{r,r,i}$, was $exp\left(-{\left\|r_{i},{\hat{r}}_{r,i}\right\|}_{2}/\sigma_{r,i\;}\right)$. Similarly, the similarity between the DNA methylation-based prediction and the observed mRNA profile for patient i, denoted as $\theta_{r,m,i}$, was calculated as $exp\left(-{\left\|r_{i},{\hat{r}}_{m,i}\right\|}_{2}/\sigma_{r,i\;}\right)$. Here, the term ${\left\|\cdot \right\|}_{2}$ represents the L2 norm, and $\sigma_{r,i\;}i$s the bandwidth for patient i, calculated as the mean Euclidean distance between patient i’s mRNA profile $r_{i}$ and those of its k nearest patient neighbors. For each omics type, we calculated the ratio of within-omics prediction similarity to the cross-omics prediction similarity. The resulting ratios were then processed using the SoftMax function to generate weighs, which was subsequently used for affinity matrix integration. Taking the mRNA profile of patient i as an example, the weight assigned was calculated as follows:$$w_{r,i}=\frac{\exp s_{r,i}}{\exp s_{r,i}+\exp s_{m,i}},$$
where $s_{r,i}=\frac{\theta_{r,r,i}}{\theta_{r,m,i}+\epsilon }$ and $s_{m,i}=\frac{\theta_{m,m,i}}{\theta_{m,r,i}+\epsilon }$, and ϵ is a small positive constant to prevent division by zero. The weight assigned to the DNA methylation affinity matrix was calculated analogously. Finally, the integrated affinity matrix can be calculated as $A=[a_{i,j}]={[w}_{r,i}a_{r,\;i,j}+w_{m,i}a_{m,\;i,j}].$ To ensure comparability across different affinity values, the integrated matrix was normalized using the Min-Max method. These weights for each patient, which reflect the relative contribution of each omics layer to the overall similarity matrix, enhance interpretability and precision.

Based on the final normalized integrated affinity matrix, patients were clustered into different subtypes using the consensus clustering method, which aggregates results from multiple iterations of hierarchical clustering to derive a more stable and robust clustering outcome [36]. Consensus clustering offers significant advantages in terms of stability, robustness, and reduced sensitivity to parameter selection, particularly when handling complex datasets or when higher clustering stability is required [36].

### 4.3. Selection of the Number of Clusters

Determining the optimal number of clusters is a challenging problem in clustering analysis, as different cluster numbers may lead to varying clustering results [37]. In this study, we evaluated cluster numbers from 2 to 10 and performed clustering analyses for each tumor. The optimal number of clusters was selected based on Silhouette values [38], with the cluster number that yields the highest Silhouette Score selected as the optimal choice. Alternatively, users can specify a cluster number based on their prior knowledge or experience.

### 4.4. Clustering Validation Metrics

To evaluate the clustering quality, we employed two widely recognized validation metrics: the Davies–Bouldin Index (DBI) [39] and the Calinski–Harabasz Index (CHI) [40].

The DBI measures the average similarity ratio between each cluster and the cluster most similar to it, considering both the within-cluster scatter and the separation between clusters. It is calculated as follows:$$\mathrm{DBI}=\frac{1}{n}\sum_{i=1}^{n}{max}_{j\neq i}\left(\frac{a_{i}+a_{j}}{d_{ij}}\right)$$
where n is the number of clusters, $a_{i}$ and $a_{j}$ are the average distances of all points in cluster i and cluster j to their respective centroids, and $d_{ij}$ is the distance between the centroids of cluster i and cluster j. Lower DBI values indicate better clustering performance, characterized by more compact and well-separated clusters.

The CHI, also known as the Variance Ratio Criterion, assesses the ratio of the between-cluster dispersion to the within-cluster dispersion, defined by$$\mathrm{CHI}=\;\frac{T_{r}(B_{k})/\left(k-1\right)}{T_{r}(W_{k})/(N-k)}$$
where k is the number of clusters, N is the total number of data points, $T_{r}\left(B_{k}\right)$ represents the trace of the between-cluster dispersion matrix, and $T_{r}(W_{k})$ denotes the trace of the within-cluster dispersion matrix. Higher CHI values suggest well-defined and distinct clusters.

### 4.5. Prognosis Analysis

Survival analyses were conducted using the R packages “survminer” and “survival”. Based on the overall survival time from TCGA clinical data, we compared the survival rates of different subtypes and plotted the survival curves. A log-rank test was carried out to compare the survival distribution differences. *p*-values less than 0.05 were considered to indicate a significant difference in the survival rate across different subtypes.

### 4.6. Immune Cell Gene Signature Scoring

Gene signature scores were computed using the code and methodology provided in the study by Alexis J. Combes et al. in *Cell* (2022) titled “Discovering Dominant Tumor Immune Archetypes in a Pan-Cancer Census” [22]. The approach involves normalizing gene expression data using the trimmed mean of M-values (TMM) method [41] to account for differences in the sequencing depth across samples.

The normalized log2CPM (log2 counts per million) expression values were arranged into an m*n matrix, where m represents the feature signature genes and n represents the selected sample set. The expression of each gene was then converted into percentile ranks across the samples using the SciPy (version 1.11.4) [42] Python module.

The final score for each sample was calculated as the percentile of the average percentile rank of the feature signature genes within that sample. The code provided by Alexis J. Combes et al. (2022) [22] was used directly without modification for this analysis.

### 4.7. Gene Set Variation Analysis

In order to compute the single-sample gene set enrichment, we utilized the Gene Set Variation Analysis (GSVA) program [43] with log2 (FPKM+1)-transformed expression data as the input. This allowed us to derive the absolute enrichment scores of the Cancer Hallmark gene set collections found in MSigDB [31]. Consequently, we were able to assess the enrichment score for each dataset and signature.

### 4.8. Drug Recommendation Analysis

We collected all the targeted drugs used to treat kidney cancer from the Drugbank database [32], as well as their target genes. Only the drugs with the mechanism of action as an inhibitor or antagonist were considered, resulting in 54 drug–gene pairs consisting of 12 drugs and 27 target genes. The differences in target gene expression were evaluated using the Wilcoxon rank-sum test. *p*-values less than 0.05 were considered significant.

## 5. Conclusions

In conclusion, we presented EMitool, a robust and explainable method for multi-omics data integration specifically tailored to disease subtyping. The subtypes generated using EMitool enable a range of downstream analyses, including a prognosis analysis, biomarker identification, and drug recommendation. This capability highlights EMitool’s potential to significantly advance precision medicine.

## Figures and Tables

**Figure 1 ijms-26-04268-f001:**
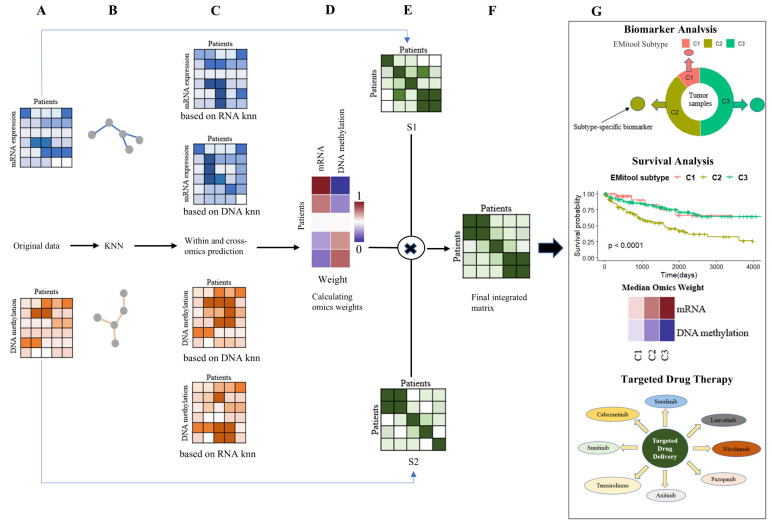
EMitool workflow. (**A**) Example representation of mRNA expression and DNA methylation datasets for the same cohort of patients. (**B**) Constructing independent k-nearest neighbor (KNN) graphs. (**C**) Performing within and cross-omics prediction. (**D**) mRNA and DNA methylation omics weights for all patients in the dataset. (**E**) Patient-by-patient similarity matrices for each data type. (**F**) The final integration matrix was clustered into different subtypes. (**G**) Downstream analysis: survival analysis curves of EMitool subtype and the median mRNA and DNA methylation omics weights for each EMitool subtype. Subtype-specific biomarker analysis. Drug response analysis for the EMitool subtype.

**Figure 2 ijms-26-04268-f002:**
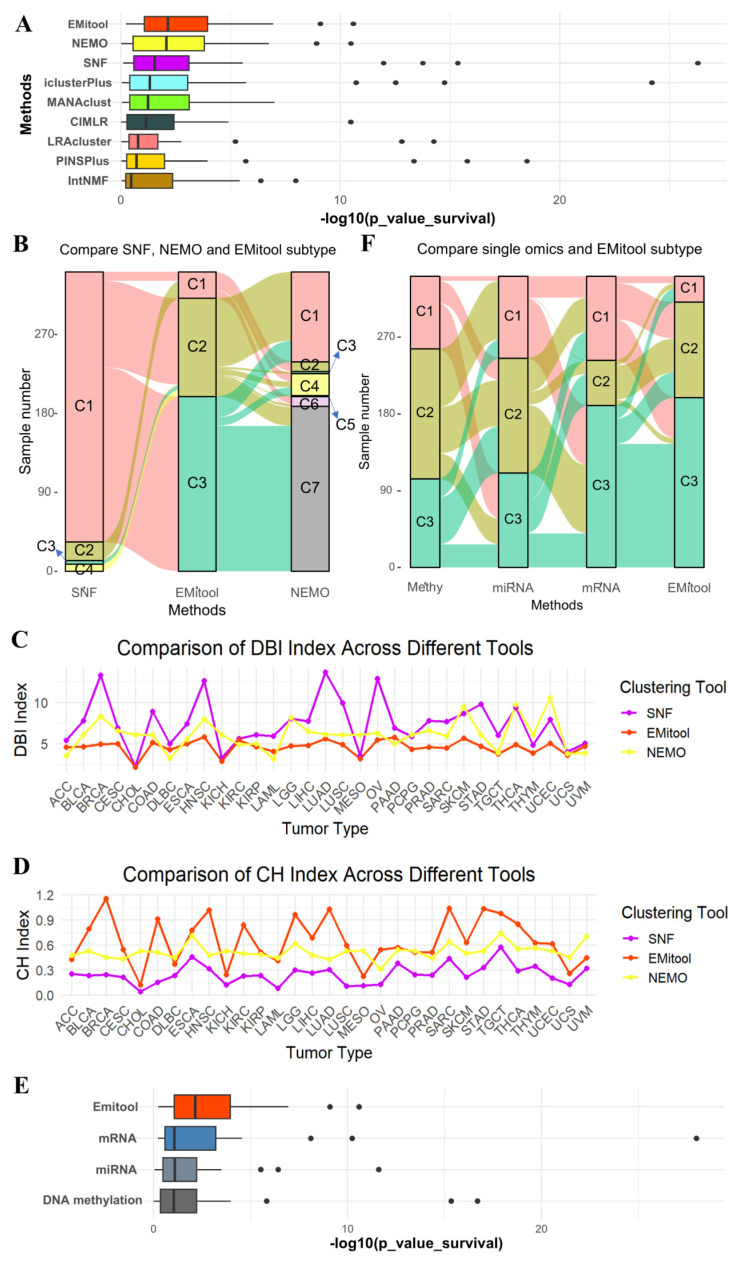
Performance comparison between EMitool and other state-of-the-art methods. (**A**) Box plot showed the −log10 (Cox log-rank test *p* value for survival analysis) for 31 cancer types. (**B**) The Sankey chart shows the difference in the classification of Kidney Renal Clear Cell Carcinoma samples using SNF, NEMO, and EMitool. (**C**) A comparison of the DBI for different tools across 31 tumor types was conducted using a line chart. (**D**) A comparison of the CHI for different tools across 31 tumor types was conducted using a line chart. (**E**) Comparison of EMitool approach and single-omics data clustering results based on TCGA data. Box plot displaying the −log10 (Cox log-rank test *p* value for survival analysis) for 31 cancer types. (**F**) The Sankey chart shows the difference in the classification of Kidney Renal Clear Cell Carcinoma samples based on single-omics and EMitool.

**Figure 3 ijms-26-04268-f003:**
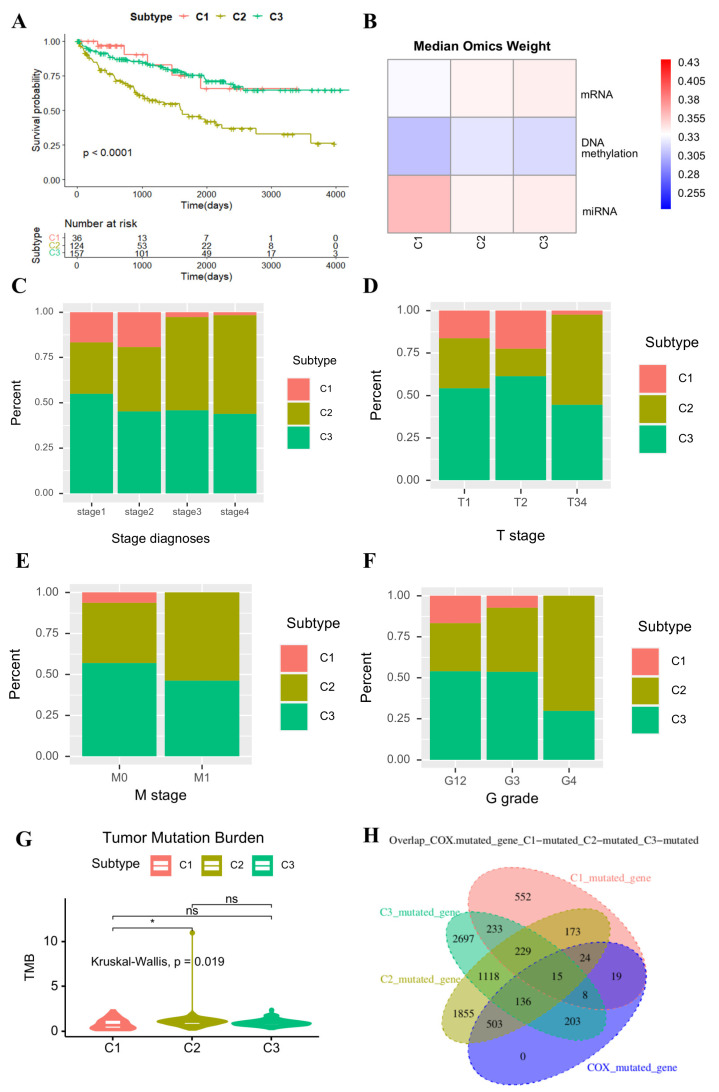
The association between EMitool subtypes and clinical features. (**A**) Kaplan–Meier survival curve for KIRC EMitool subtype. (**B**) The weight of the different omics in each EMitool subtype. (**C**) The proportion of patients with different EMitool subtypes in tumor stage diagnoses. (**D**) The proportion of patients with different EMitool subtypes based on T stages (T34 represents the combination of T3 and T4 stages, because T3 and T4 have too few patients). (**E**) The proportion of patients with different EMitool subtypes based on M stages. (**F**) The proportion of patients with different EMitool subtypes based on histologic grade (G12 represents the combination of G1 and G2 grade, because G1 and G2 have too few patients). (**G**) The boxplot represents Tumor Mutation Burden for KIRC EMitool subtype (the Wilcoxon rank-sum test was used; ns *p* > 0.05, * *p* < 0.05). (**H**) Intersection of KIRC prognosis-related mutated genes with mutated genes of each EMitool subtype.

**Figure 4 ijms-26-04268-f004:**
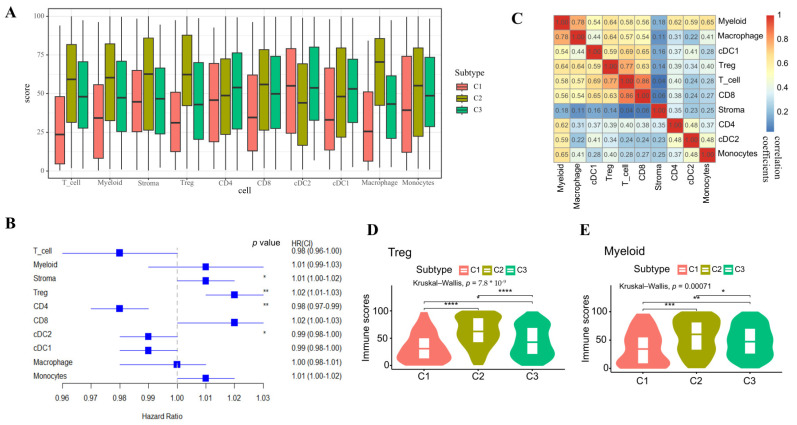
Immune microenvironment analysis. (**A**) The boxplot shows the proportion of 10 types of immune cells based on the KIRC EMitool subtype. (**B**) The forest plot indicates the HR (squares) and 95% CI (whiskers) for overall survival (KIRC patients) calculated using a multivariate Cox proportional hazard regression model based on immune cells, as indicated; *p* values < 0.05 are marked with an asterisk (* *p*  <  0.05, ** *p*  <  0.01). (**C**) The heatmap represents the Pearson correlation analysis between cell proportions. The numbers in the heatmap indicate the correlation coefficients. (**D**,**E**) Comparison of Treg and myeloid cell scores among different subtypes (the Wilcoxon rank-sum test was used; ns *p* > 0.05, * *p* < 0.05, ** *p* < 0.01, *** *p* < 0.001, **** *p* < 0.0001).

**Figure 5 ijms-26-04268-f005:**
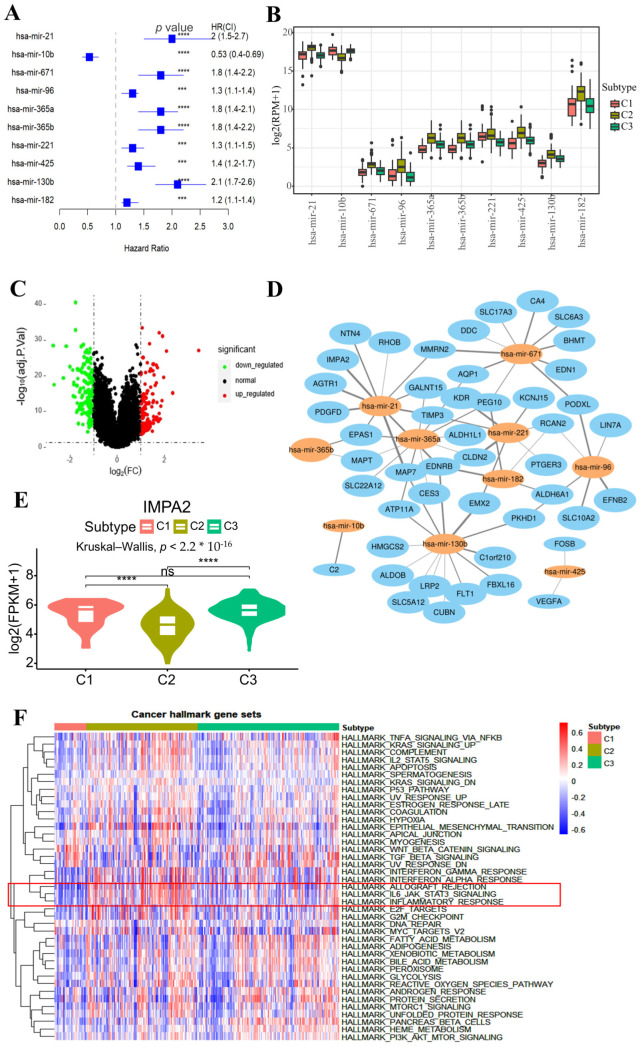
Identification of biomarkers for subtype C2. (**A**) The forest plot indicates the HR (squares) and 95% CI (whiskers) for overall survival (KIRC patients) calculated using a univariate Cox proportional hazard regression model based on miRNA expression, as indicated; *p* values < 0.05 are marked with asterisk (*** *p* < 0.001, **** *p* < 0.0001). (**B**) A boxplot illustrating the expression comparison of the top 10 miRNAs across different subtypes. (**C**) A volcano plot depicting the differential gene expression comparison between subtype C2 and other subtypes. (**D**) A regulatory network diagram depicting miRNA–gene interactions. Orange nodes represent miRNAs, blue nodes represent genes, and edges represent Pearson correlation coefficients between miRNAs and genes. Thicker edges indicate stronger regulatory effects of miRNAs on genes. (**E**) The comparison of IMPA2 gene expression between different subtypes was performed (Wilcoxon rank-sum test was used. ns *p* > 0.05, **** *p* < 0.0001). (**F**) The heatmap displays the GSVA enrichment scores of different subtypes in the cancer hallmark gene set.

## Data Availability

An open-source implementation of EMitool in R is available from https://github.com/megasci-lab/EMitool (accessed on 2 April 2025). The R package is available at https://api.github.com/repos/megasci-lab/EMitool/tarball/HEAD (accessed on 2 April 2025).

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
