# Peer review of "EMitool: Explainable Multi-Omics Integration for Disease Subtyping"

_ijms, 2025, doi:10.3390/ijms26094268_

Round 1
Reviewer 1 Report
Comments and Suggestions for Authors
Dear authors,
The manuscript prepared by Xu et al. provides an interesting integration tool for use in multi-omics, possibly providing a solution to high-speed data accumulation worldwide. It has clearly stated translational potential; it is a novel approach in disease subtyping. The necessary comparison with other advanced, well-known methods in various cancers was presented. The most promising subtyping methods were compared with EMitool, which outperformed SNF and NEMO in following clustering. The manuscript is very well written, interesting, and focused on integration tools for omics, which is a promising approach for the future of science based on generated data. It was proven as robust. I hope the limitations indicated by the authors will be studied for further improvement of the tool, but there is an excellent approach of providing the tool on github as an open-source R package. The visualization of data is great, figures are attractive for a reader and informative. Hopefully, the authors will extend the discussion further by indicating the limitations. The possible uses in the generation of omics data, considering others work in the area, besides transcriptomics. Do you consider it useful in proteomics and metabolomics also?
Reviewer 2 Report
Comments and Suggestions for Authors
In this manuscript, the authors introduce a novel multi-omics integration approach designed to facilitate explainable disease subtyping. Overall, this manuscript requires major revisions to further improve its quality. Specific comments are given below:
- In the introduction section, the motivation and novelty of this manuscript should be further strengthened. Specifically:
(1) The analysis of the limitations of existing multi-omics integration techniques is somewhat general and lacks an in-depth comparison of the advantages and disadvantages of different methods in practical applications. This may prevent readers from gaining a comprehensive understanding of the current research status in the field, thus affecting the demonstration of the manuscript's novelty.
(2) The authors highlight that an important advantage of EMitool is its interpretability, but there is a lack of in-depth discussion on the specific implementation mechanisms and technical details of EMitool's interpretability.
- When introducing data preprocessing and normalization, the authors mention that for the IntNMF method, the original level-three data format is used directly without applying standard normalization methods. It is unclear whether this preprocessing approach could lead to ineffective adjustment of dimensional and distribution differences between different omics data, potentially affecting the accuracy of subsequent data integration and analysis. The authors should provide a more detailed explanation of this.
- In calculating the similarity matrix, the authors generate a comprehensive similarity matrix by calculating the similarity between the observed and predicted values (within- or cross-omics). However, the authors only approach similarity calculation from a mathematical perspective and do not explore the biological basis of these similarity values.
- When comparing the performance of EMitool with eight other advanced methods in cancer subtyping, the authors primarily evaluate based on survival analysis significance, DBI, and CHI indices. While this comparison can reflect the performance advantages of EMitool to some extent, it lacks an in-depth comparison of the biological significance of disease subtypes and the clinical application value of different methods.
- In future work, the authors may consider incorporating more advanced deep learning techniques, such as graph representation learning (10.1109/TNSE.2024.3524077, 10.1109/JBHI. 2024.3357979) to further enhance the predictive power of models in this field.
Round 2
Reviewer 2 Report
Comments and Suggestions for Authors
All of my concerns have been addressed in this revision.